# Comparison of Single Target-Controlled Infusion Pump-Delivered Mixed Propofol and Remifentanil with Two Target-Controlled Infusion Pumps-Delivered Propofol and Remifentanil in Patients Undergoing Breast Cancer Surgery—A Prospective Study

**DOI:** 10.3390/ijerph20032094

**Published:** 2023-01-23

**Authors:** Hou-Chuan Lai, Meng-Fu Lai, Yi-Hsuan Huang, Jyh-Cherng Yu, Wei-Cheng Tseng, Zhi-Fu Wu

**Affiliations:** 1Department of Anesthesiology, Tri-Service General Hospital and National Defense Medical Center, Taipei 11490, Taiwan; 2Division of General Surgery, Department of Surgery, Tri-Service General Hospital and National Defense Medical Center, Taipei 11490, Taiwan; 3Department of Anesthesiology, Kaohsiung Medical University Hospital, Kaohsiung Medical University, Kaohsiung 80756, Taiwan; 4Department of Anesthesiology, Faculty of Medicine, College of Medicine, Kaohsiung Medical University, Kaohsiung 80708, Taiwan; 5Center for Regional Anesthesia and Pain Medicine, Wan Fang Hospital, Taipei Medical University, Taipei 11696, Taiwan

**Keywords:** remifentanil, propofol, total intravenous anesthesia, breast cancer surgery

## Abstract

Total intravenous anesthesia (TIVA) with remifentanil and propofol (RP) is considered to be an ideal type of general anesthesia (GA) for pediatric and adult patients undergoing medical procedures. However, delivery of an RP mixture by target-controlled infusion (TCI) for GA in surgical procedures has not been described. We investigated the merit of this approach for breast cancer surgery. Eighty-four patients (n = 42 per group) were randomly allocated to propofol and remifentanil either delivered by separate TCI pumps (S group) or in an RP mixture by a single TCI pump (M group). Dosages were adjusted based on the bispectral index (BIS) and the analgesia nociception index (ANI). The primary outcomes were adequate anesthesia (BIS 40–60 and ANI 50–70, respectively), acceptable hemodynamic fluctuations (<30% of baseline) with less frequent TCI pump adjustments, bolus injections of anesthetics, and total consumption of anesthetics during the procedure. The secondary endpoints included time of emergence from anesthesia, patient satisfaction, postoperative pain, rescue with opioids, and adverse events. The characteristics of patients, hemodynamic parameters, BIS and ANI scores, duration of surgery, anesthesia, and emergence were not significantly different between groups. The adjustment frequency of TCI was significantly higher in the S group (3 (range 0–6) vs. 2 (0–6) times; *p* = 0.005). The total dosage of anesthetics, pain rating, patient satisfaction, need for opioids postoperatively, and incidence of adverse events were not significantly different. We have demonstrated that this RP mixture provided adequate hypnotic and analgesic effects under BIS and ANI monitoring in patients undergoing breast cancer surgery within 1 h.

## 1. Introduction

The use of remifentanil is preferable to other opioids for anesthesia when combined with propofol in target-controlled infusion (TCI) regimens for its unique properties: rapid onset of action, precise intra-operative control, and a fast recovery profile [1,2]. Propofol and remifentanil are often administered with two TCI pumps for providing hypnotic and analgesic effects, respectively, and are considered to be ideal anesthetic techniques [3]. In addition to a doubling in the need for TCI supplies (such as pumps, syringes, and extension tubes), it is more time-consuming to set TCI models for propofol and remifentanil when these are being infused separately. A remifentanil–propofol (RP) mixture in the same syringe for total intravenous anesthesia (TIVA) delivery has been reported to be safe and effective in procedural sedation and general anesthesia (GA) [4,5,6,7,8,9,10,11,12,13,14,15,16,17,18]. Research on the use of an RP mixture for maintenance of GA is limited [16,17,18]. Brady et al. concluded that remifentanil (25 mcg.mL^−1^) in 0.5% propofol was more cost-effective than conventional balanced anesthetics, resulting in quicker patient discharge, and has important implications for ambulatory surgery centers and office-based practices [16]. Bagshaw et al. concluded that using a mixture of remifentanil (5 mcg.mL^−1^) in 1% propofol was associated with a lower incidence of adverse events in 873 pediatric patients with TIVA [17]. In addition, Malherbe and Barker performed at least 7000 procedures per year with a ‘PR5’ mixture (propofol 1% in combination with remifentanil, 5 mcg.mL^−1^). Their clinical experience has been similar to the research by Bagshaw et al. [17]: the incidence of adverse events is very low [19]. Bakan et al. reported that the infusion of remifentanil (20 or 30 mcg.mL^−1^) in 1% propofol for thyroidectomies did not result in any statistically significant difference in recovery or clinical outcomes compared with the standard TIVA technique using separate drug infusions [18]. 

Stewart et al., reported that remifentanil can be mixed with propofol at low concentrations (5 mcg.mL^−1^), giving a remaining remifentanil concentration of 91.9% after 1 h [20]. Eleveld et al., showed that the flow rate of the TCI pump was decreased over time to maintain the target effect-site concentration (Ce) of propofol [21]. Due to the instability of the RP mixture, and infusion rates decrease over time to maintain a stable blood concentration, as the context-sensitive half-time rises with the TCI of propofol’s Ce; therefore, the rate of remifentanil infusion will decrease, resulting in lower concentrations that might not be enough for surgery [22]. Thus, the RP mixture is prepared just before anesthesia induction for short surgical procedures, and low concentrations of remifentanil and monitoring the depth of anesthesia and analgesia are necessary.

The analgesia nociception index (ANI) is an index that measures the high-frequency component of heart-rate variability (parasympathetic component of the autonomic response) using the electrocardiogram signal as an input on a scale from 0 (maximum of nociception) to 100 (complete analgesia) [23,24,25,26]. Based on the published studies [23,24,25,26], an ANI of between 50 and 70 may correspond to adequate antinociception. Gruenewald et al. reported that ANI enabled consistent reflection of stimulation during propofol-remifentanil anesthesia [23]. The ANI is superior for detecting painful stimulations compared to heart rate and mean arterial pressure during propofol and remifentanil anesthesia [24]. In breast cancer patients under GA, use of ANI monitoring can help optimize opioid consumption and provide data about nociception/antinociception intraoperatively [25]. Recently, in a randomized controlled trial, Sabourdin et al. demonstrated that ANI guidance resulted in lower remifentanil consumption compared with standard practice under propofol and remifentanil anesthesia [26]. In previous clinical studies of RP mixtures, their monitors only included the hemodynamic monitors and sometimes a bispectral index (BIS) monitor, and there was no ANI monitor. Here, we first introduce ANI-guided analgesia for RP mixtures.

In our institution, even in Taiwan, multiple TCI systems are not popular in each operating room (usually only a single TCI pump available). Considering the limited effective time for RP mixtures and breast cancer surgery typically taking less than 1 h in our hospital when performed by an experienced surgeon, this study aimed to assess the efficacy of the RP mixture prospectively for TCI of propofol’s Ce in patients undergoing breast cancer surgery under BIS and ANI monitoring.

## 2. Materials and Methods

### 2.1. Study Design and Setting

This was a prospective study conducted at the Tri-Service General Hospital (Taipei, Taiwan).

### 2.2. Participants and Data Sources

This was a randomized, interventional clinical study that enrolled patients undergoing breast cancer surgery between August 1, 2018, and December 31, 2018, in Tri-Service General Hospital, Taipei, Taiwan. The study protocol was approved by the Institutional Review Board of our hospital (TSGH IRB No: 2–107-05-079) and registered in the ClinicalTrial.gov database (NCT03817359). Participants were recruited with written informed consent obtained in the ward the day before surgery, after being provided with detailed information. Inclusion criteria were American Society of Anesthesiologists (ASA) grades II to III, age >18 y, and being scheduled for breast cancer surgery under TIVA. Exclusion criteria included previously determined allergic reactions to propofol or remifentanil, and the performance of other surgical procedure in the same session leading to extended operative time of > 1 h. Patients were randomly and equally assigned to two groups (separate, S group; mixed, M group) using computer-generated random numbers. All operations were conducted by the same surgeon (JC Yu), and all anesthesia was performed by one single anesthesiologist (ZF Wu). The data were recorded by a nurse anesthetist who was blinded. In addition, cases and the operator were blinded. 

### 2.3. Mixture Preparation

Remifentanil powder (2 mg) was reconstituted with 2 mL water for injection to a final concentration of 1 mg.mL^−1^. The reconstituted remifentanil solution (0.25 mL; 250 mcg) was mixed in a bottle of propofol (1%; 10 mg.mL^−1^), 50 mL (Fresofol 1% MCT/LCT 50 mL), and aspirated in a 50 mL polypropylene syringe, giving final concentrations of remifentanil and propofol of 5 mcg.mL^−1^ and 10 mg.mL^−1^, respectively. Each mixture was inspected visually before anesthesia induction to ensure no layering or separation of the constituents.

### 2.4. Anesthesia and Monitoring

All patients fasted overnight before the procedure, and no medications were allowed before the induction of anesthesia. Standard monitoring, such as electrocardiography (lead II), non-invasive blood pressure (NIBP), pulse oximetry oxygen saturation, end-tidal carbonic dioxide (ETCO_2_), and train-of-four (TOF) were used for all patients. In addition, all patients underwent monitoring for the BIS (BIS VISTA™, Aspect Medical Systems, Inc. Norwood, USA) and ANI (Physiodoloris^®^, MDoloris Medical Systems, Loos, France). Participants were preoxgenated with 6 L.min^−1^ 100% oxygen via a facial mask to achieve peripheral oxygen saturation of 99–100% before induction.

### 2.5. Induction and Maintenance

In the S group, patients were induced with an effect-site concentration (Ce) of 2.0 ng.mL^−1^ of remifentanil (50 mcg.mL^−1^, Minto model) and Ce 4.0 mcg.mL^−1^ of propofol (10 mg.mL^−1^; Schnider model) with continuous infusion using two separate TCI pumps (TCI, Fresenius Orchestra Primea; Fresenius Kabi AG, Bad Homburg, Germany). In the M group, the induction was performed using an RP mixed regimen using single-TCI-pump infusion with the Schnider model of propofol Ce 4.0 mcg.mL^−1^, but not the Eleveld model based on our routine practice [7]. Rocuronium (0.6 mg.kg^−1^) was administered after loss of consciousness to all patients to facilitate endotracheal intubation using a Trachway video stylet (Biotronic Instrument Enterprise Ltd., Tai-Chung, Taiwan).

Anesthesia was maintained with an oxygen flow of 0.3 L.min^−1^ and mixed air 0.3 L.min^−1^. The ETCO_2_ was maintained at 35–45 mm Hg by adjusting the ventilation rate and maximum airway pressure to < 30 cmH_2_O. Maintenance doses of rocuronium (0.1 mg.kg^−1^) were given at TOF count ≥ 1 throughout the operation.

The intraoperative administration of propofol and remifentanil was guided by maintaining the BIS value at 40–60 and a keeping 4 min moving average ANI of 50–70 during surgery. For the S group, the Ce of remifentanil was kept at 2.0 ng.mL^−1^, and the Ce of propofol was decreased to 2.5 mcg.mL^−1^ if the hemodynamic parameters remained stable after endotracheal intubation. The Ce of remifentanil was then adjusted in steps of 0.2 ng.mL^−1^ based on ANI monitoring, and the Ce of propofol was adjusted in steps of 0.2 mcg.mL^−1^ guided by BIS monitoring during the surgical procedure.

For the M group, the Ce of the RP mixture regimen was decreased to 2.5 mcg.mL^−1^ after endotracheal intubation. The protocols for intraoperative adjustment were as follows: (1) BIS > 60 and ANI 50–70 or >70 with a bolus injection of propofol 30 mg repeated every 3 min until the BIS score decreased below 60; (2) for BIS > 60 and ANI <50, we increased the RP mixture regimen Ce by 0.2 mcg.mL^−1^ and repeated this every 3 min until the BIS decreased below 60 and ANI scores increased above 50; (3) for BIS 40–60 and ANI <50, we gave bolus injections of remifentanil 12.5 mcg and repeated this every 3 min until the ANI score increased above 50; (4) for BIS < 40 or 40–60 and ANI > 70, we reduced the RP mixture regimen by 0.2 mcg.mL^−1^ and repeated this every 3 min until the BIS was 40–60 and the ANI was 50–70; (5) for BIS < 40 and ANI 50–70, we reduced an RP mixture regimen by 0.2 mcg.mL^−1^ every 3 min until the BIS was 40–60 and ANI between 50–70; (6) for BIS < 40 and ANI < 50, we reduced the RP mixture regimen by 0.2 mcg.mL^−1^ and gave a bolus remifentanil of 12.5 mcg and repeated this every 3 min until the BIS was 40–60 and ANI scores were > 50. This bolus regimen was conservative as a precaution but consistent with clinical practice at our institution. These 3 min assessment periods coincided with the automatic BP measurement intervals and allowed for an 80 to 120 s delay in ANI to maximally reflect nociceptive stimulation [23]. For patients with elevations or drops in heart rate (HR) and blood pressure (BP) (>30% of baseline) during anesthesia, nicardipine or ephedrine were administered based on the hemodynamic status [27].

### 2.6. Emergence from Anesthesia

The propofol infusion was stopped after skin closure, and the Ce of remifentanil was maintained as 1.0 ng.mL^−1^ while patients were emerging from anesthesia to prevent remifentanil-induced hyperalgesia (RIH) in the S group [28]. In the M group, infusion of the RP mixture was stopped after skin closure. All patients received ketorolac (30 mg intravenously) prior to wound closure and local anesthetic wound infiltration with lidocaine for postoperative pain management. Reversal of neuromuscular function was achieved by administering sugammadex (2–4 mg.kg^−1^) to prevent residual paralysis. When the patient regained consciousness with spontaneous and smooth respiration (TOF ratio ≥ 0.9), the endotracheal tube was removed, and the patient was sent to the post-anesthesia care unit (PACU) for further care. An independent investigator recorded the values of BIS and ANI, and the frequency of TCI device adjustments. Any bolus of remifentanil or propofol given intraoperatively was defined as a TCI pump adjustment. The hemodynamic parameters, including mean blood pressure (MBP) and HR, were recorded at baseline and at the following times: after intubation; at surgical incision; 30 min after skin incision; at the end of surgery; after extubation; and at emergence from anesthesia. Postoperative pain was assessed subjectively by the patients using a visual analogue scale (VAS) on arriving and leaving the PACU (the first postoperative hour) and on the first postoperative day [29]. When the patient’s VAS was > 4, fentanyl, 50 mcg, was given intravenously in the PACU. In this study, the RIH was defined as VAS > 4 after prescribing fentanyl ≥ 50 mcg within the first postoperative hour. The various time intervals (surgical time, anesthetic time, time for emergence); postoperative rescue analgesia; and adverse effects, such as postoperative nausea and vomiting (PONV) and RIH, were retrieved through anesthetic records and electronic medical records retrospectively. The duration of each phase was defined as follows: (1) surgical time: from skin incision to skin closure; (2) anesthetic time: from the beginning of induction to extubation; (3) time for emergence: from the end of surgery to eye-opening on verbal command; (4) overall patients’ satisfaction on postoperative day one was rated by a questionnaire using the following scale: 1, very unsatisfactory; 2, unsatisfactory; 3, neutral; 4, satisfactory; and 5, very satisfactory.

### 2.7. Outcomes

The primary aim of this study was to determine which anesthetic technique provided adequate anesthesia (BIS 40–60 and ANI 50–70) and acceptable hemodynamic fluctuations (<30% of baseline) with less TCI pump adjustments, bolus injections of anesthetics, and total consumption of anesthetics during the procedure. Secondary outcome measures included time of emergence from anesthesia, patient satisfaction, postoperative pain, rescue with opioids, and adverse events.

### 2.8. Statistical Analysis

Data are expressed as mean ± standard deviation (SD), as median and range, or as number of cases and percentage. Demographic and perioperative variables were compared using Student’s t-test, or the Mann–Whitney nonparametric U test if the data were not normally distributed. Categorical variables were compared using the Chi-squared test or Fisher’s exact test if need. Statistical significance was accepted for two-tailed *p* values of <0.05. The statistics were calculated by using SigmaPlot for Windows version 14.5 (Systat Software Inc, CA, USA).

### 2.9. Power and Sample Size

Based on the same surgical population (200 cases) in our institution, a power analysis was performed using the frequencies of adjustment of propofol and remifentanil doses during surgery as the primary variables, where the mean and SD of adjustment times were set to 3.0 and 1.1, respectively. A power analysis indicated that at least 42 patients were required in each group to detect a difference of 0.6 points in the VAS score (20% of the non-inferiority margin selected for clinical consideration), with a type I error of 0.05 and a power of 80%. With an estimated 10% dropout rate, we enrolled 92 patients. Ultimately, 84 patients were included for analysis.

## 3. Results

### 3.1. Baseline Characteristics

Ultimately, 92 patients were recruited and 84 were analyzed. Three patients were excluded for having concomitant breast reconstruction surgery, and five participants for incomplete ANI and BIS data caused by a temporary technical failure intraoperatively, leaving 84 patients included for analysis (Figure 1).

No significant differences were found between the two groups with respect to demographic factors and the times of operation, anesthesia, and emergence from anesthesia (Table 1).

### 3.2. Perioperative Parameters for the Two Groups

Hemodynamic profiles at baseline, intubation, skin incision, 30 min after surgical incision, and extubation were all similar in both groups (Figure 2). The total instances of TCI device adjustment were significantly fewer in the M group compared with the S group (2 (range 0–6) vs. 3 (0–6); *p* = 0.005) (Table 2). There was no significant difference in the instances of Ce increase by TCI pump adjustments between groups. However, the instances of Ce reductions of the TCI pump settings intraoperatively were significantly fewer in the M group (1 (0–4) vs. 2 (0–4); *p* = 0.016). In addition, the median numbers of bolus injections of propofol and remifentanil were 0 (0–1) and 0 (0–1) in the M group. Overall, there were no significant differences in the total infused dosages of propofol (S group 437.0 ± 92.2 vs. M group 454.3 ± 87.6 mg; *p* = 0.380) or remifentanil (S group 240.6 ± 71.9 vs. M group 228.5 ± 43.3 mcg; *p* = 0.353). There were no significant differences in the ANI (Figure 3) and BIS (Figure 4) scores, in the numbers of patients requiring rescue analgesia postoperatively (12 vs. 14; *p* = 0.814), in the numbers of PONV (0 vs. 1; *p* = 1.000), or in patient satisfaction (Table 2). In the PACU, there was no patient with RIH in either group. Finally, there was no nicardipine or ephedrine use recorded in either group.

## 4. Discussion

The major finding in this study was that remifentanil combined with propofol proved a safe and effective anesthetic approach in breast cancer surgery under BIS and ANI monitoring. Our result was consistent with previous studies showing that the RP combination is practical during not only procedure sedation, such as for fibreoptic bronchoscopy, MRI, colonoscopy, breast biopsy surgery, shock wave lithotripsy, day-case urological surgery, radiation (photon beam) therapy, cataract surgery, endoscopic retrograde cholangiopancreatography, and third molar surgery [4,5,6,7,8,9,10,11,12,13,14,15], but also for GA [16,17,18]. In addition, our study provides evidence that an RP mixture in breast cancer surgery under BIS and ANI monitoring is an alternative for double-pump TCI. Further larger patient populations and surgical types are warranted to evaluate this approach. We summarize the use of an RP mixture for anesthesia in clinical or laboratory studies in Appendix A.

TCI with propofol is an ideal anesthetic technique with advantages in terms of less PONV, a rapid patient awakening profile, and helping to minimize malignant hyperthermia [3,30]. The combination of propofol with other anesthetic drugs is a common practice for delivering TIVA, mostly administered simultaneously with separate TCI systems. Under certain circumstances where multiple TCI systems are unavailable or time- and cost-consuming, mixing multiple drugs in a single syringe is a feasible and well-established alternative, including propofol with ketamine (known as ‘ketofol’), propofol with alfentanil, or even a 3-in-1 regimen combining propofol, alfentanil, and mivacurium by intravenous bolus or infusion by syringe infusion pumps [31,32,33,34,35]. However, this limited clinical use is largely attributed to previous reports of the instability of RP mixtures [20,36]. Fixed drug proportions does not allow control of individual pharmacodynamic (PD) effects, undermining many of the advantages of TIVA [22]. Nestor et al., argued that the main problems with RP mixtures were the PK and PD interactions of these drugs and the ascertainment of physical compatibility. They emphatically discouraged novices from beginning by using a drug mixture that is not licensed and using a PK model that it had not been designed for [22]. Absalom et al., considered that mixing propofol and remifentanil creates a new unlicensed drug, so that the person mixing it takes on the responsibilities of a manufacturer. If a patient receiving anesthesia in the form of a mixed PR infusion suffers a critical incident or actual harm, the clinician’s practice may come under scrutiny and criticism, potentially involving a legal challenge [37]. In addition, O’Connor et al. have studied the interaction between propofol and remifentanil when mixed in the same syringe and suggested that they can undergo separation and layering, resulting in larger oil globules, giving a ‘theoretical’ risk of fat embolism. Therefore, they concluded that TIVA should always be delivered with drugs in separate TCI devices. However, in their study, the remifentanil concentrations of the RP mixture in the three tested groups were 25, 50, and 100 mcg.mL^−1^, which are significantly greater than those for clinical routine practice [36]. Henkel et al. reported that there were no obvious visual incompatibilities or signs of emulsion instability/separation when remifentanil and propofol were mixed over the time period tested [38]. There is currently no clinical case report of fat embolism caused by an RP mixture to date. 

Stewart et al., reported that remifentanil can be mixed with propofol at low concentrations (5 mcg.mL^−1^) in polypropylene syringes; the remaining remifentanil concentration was 91.9% after 1 h, and there was no mention of a loss of effect or toxicity [20]. Henkel et al. reported that the percentage of remifentanil remaining after reconstituting with water or 0.9% saline and then mixing with propofol decreased by 50% to 60% over 24 h, compared with the same remifentanil solutions in isolation. However, in the first hour, the percentages of remifentanil at 20 mcg.mL^−1^ and 1% propofol remained at 99.5 ± 2.5% and 99.6 ± 0.21% of those of the same remifentanil and propofol solutions in isolation [38]. Wylie et al. used remifentanil (5 mcg.mL^−1^) with 1% propofol for 57 min using the Paedfusor® TCI model for children weighing 10–20 kg, and there was no difference in the concentration of remifentanil in the residual samples [39]. In addition, according to information on the electronic Medicines Compendium, remifentanil has been shown to be compatible with propofol [36]. Therefore, 1% propofol and 5 mcg.mL^−1^ of remifentanil as an RP mixture in a syringe might be stable.

We addressed safety concerns and made sure that adequate amnesia and analgesia were maintained by the RP mixture. First, the mixture is prepared routinely just before anesthesia induction to minimize losses from possible breakdown of remifentanil in the mixture. Second, the syringe is placed horizontally in the TCI pump to prevent layer formation. Third, to achieve an adequate Ce at induction, propofol needs to be administered as a bolus, but this is likely to result in an excessive dose of remifentanil [40]. However, Nestor et al. stated that when using the propofol target concentration of 3.5 mcg.mL^−1^ (using the Eleveld model) as a means to infuse remifentanil (5 mcg.mL^−1^) at the same time, the peak effect site concentration that is reached after a bolus induction was around 5.5 ng.mL^−1^ in a 7-year-old patient (110 cm, 23 kg) [41]. In our previous report, the propofol target concentration was a Ce of 4.0 mcg.mL^−1^ (using the Schnider model) and the dose of remifentanil was ~30–50 mcg (Ce of 4.0–6.0 ng.mL^−1^) during induction, and excessive dosage did not occur [42]. Fourth, the propofol infusion rates delivered by a TCI pump decrease over time to maintain a stable Ce, resulting in a decrease in the rate of the remifentanil infusion. The decreased infusion rate of remifentanil might cause lower concentrations of remifentanil and might not be enough for safe surgery [22]. However, Nestor et al. stated that the combined effect of both propofol and remifentanil approximates the effective concentration to achieve the desired effect in 95% of the population using the Eleveld model [41]. To make our findings more reliable, all participants underwent BIS and ANI monitoring throughout the procedure, allowing the adjustment of anesthetic concentrations in a more precise way. The BIS correlates well with a hypnotic state and could reduce the incidence of intraoperative awareness or the need for anesthetic use. It has been certified as a standard monitoring system for TIVA [43,44,45]. Funcke et al. reported that the ANI is superior at detecting painful stimuli under sedation compared with clinical signs such as HR and MBP [24]. In a meta-analysis of randomized controlled trials, Jiao et al. reported that ANI monitoring seemed to have an advantage over standard clinical practice for the intraoperative management of analgesia during general anesthesia [46]. If there was any insufficiency in amnesia or analgesia caused by instability of the RP mixture, it would have been detected by the use of BIS and ANI monitoring in this study.

The mean anesthetic times were 67 and 68 min in the S and M groups, respectively. However, the mixed regimen infusion was stopped immediately after skin closure, and the time for emergence from anesthesia (3.9 and 2.9 min) was also included in this period. Thus, the true mixed regimen delivering period is believed to be within approximately 60 min. In addition, the BIS and ANI-guided anesthesia ensured an adequate depth of amnesia/analgesia until the end of operation. The total times of TCI adjustment in the M group were significantly lower than in the S group. The median numbers of bolus injections of propofol and remifentanil were 0 (0–1) and 0 (0–1) in the M group, respectively. This might have resulted from the ‘two-in-one’ property of the RP mixture combining both hypnotic and analgesic effects. Accordingly, when patients appear to have inadequate hypnosis or analgesia by monitoring, a single increment of Ce in the M group would achieve two goals in one step, which is as effective as adjusting individually. In addition, physicians treating the M group were inevitably confronted with a dilemma when the BIS and ANI monitors indicated opposite directions, allowing only single specific adjustment and then a ‘wait and see’ period (i.e., when BIS > 60 and ANI > 70, bolus injection of propofol for sedation and an observing period for analgesia). Thus, they were unable to alter each component separately, and the number of times of adjusting for TCI was one. However, in the same situation, the participants in the S group received increased propofol and reduced remifentanil, making two changes with respect to the Ce target.

As participants in the M group were unable to receive gradual withdrawal of remifentanil separately, this might have led to a difference in the incidence of RIH between groups [29]. To our surprise, the incidence of RIH did not increase significantly in the M group. This could be associated with the introduction of multimodal analgesia (MMA) in both groups, including wound infiltration with lidocaine and systemic administration of non-steroidal anti-inflammatory drugs, which might attenuate the incidence of RIH [2,47]. However, tapering infused opioid doses by gradual withdrawal of remifentanil in breast cancer surgery during MMA should be investigated further.

There were some limitations of this study. First, there were flaws in the methodology of this work: a lack of proper PK-PD analysis for the use of the RP mixture by a TCI pump which was specifically designed following intensive PK-PD analysis made the results less desirable and reliable. However, several clinical studies (Appendix A) showed that routine use of the RP mixture for procedural sedation or GA was safe and feasible by the syringe pump. Additionally, this was the first study to anesthetize paralyzed patients using an RP mixture by applying the Schnider model of TCI. When remifentanil is mixed with propofol and delivered as the TCI of propofol, remifentanil delivery is not target-controlled but passively follows the variable infusion rates calculated by the TCI pump to deliver predicted effect-site concentrations of propofol [37]. Thus, we strongly suggest the use of BIS and ANI monitors during surgery, and the establishment of a PK-PD model for RP mixture is much needed. Second, this was a prospective study with a relatively small sample size. In addition, the results of this study cannot be extrapolated to patients undergoing surgical procedures for more than 1 h, who are considered to be at risk of inadequate sedation/analgesia from inherent mixture instability. These strategies are commonly used in our institution;, however, a lack of significant benefit to the patients and legal challenges may become an issue [37]. Further large randomized controlled studies with complete PK and PD models to validate our findings are necessary. Third, we might meet a dilemma when the BIS and ANI indicate opposite directions for adjusting dosage, such as BIS < 40 and ANI < 50 or BIS > 60 and ANI > 70. This was the major weak point of our study, in that we could not deal with separate infusion rates. While BIS < 40 and ANI < 50, we reduced the Ce of RP mixtures and gave a bolus of remifentanil. When BIS > 60 and ANI > 70, we gave a bolus injection of propofol and checked the RIH postoperatively. In addition, we also kept hemodynamic fluctuation <30%. Furthermore, the amount of 5 mcg.mL^−1^ in 10 mg.mL^−1^ for the RP mixture that we chose was based on our clinical experience, and further investigations are required to determine the optimal RP proportions for different kinds of procedures under the monitors of the depth of anesthesia and analgesia. Fourth, RIH remains an issue of concern in TCI with a remifentanil infusion. For RP mixture regimens with a single TCI infusion, it is not possible to titrate each drug separately. Therefore, strategies for preventing RIH during MMA, such as the use of regional blockade, opioids, non-steroidal anti-inflammatory drugs, or local anesthetic infiltration into a wound are recommended. Hyperalgesia was not measured by specific devices (von Frey filaments) or pressure-threshold algometers [48]. Instead, the RIH was defined as VAS > 4 after prescribing fentanyl ≥ 50 mcg within the first postoperative hour [29]. Consistently, Jo et al. found that RIH was related to significantly higher VAS and higher consumption of rescue analgesics [49]. In addition, the gradual withdrawal combined with drip infusion of remifentanil required less rescue analgesics and reduced pain scores [28]. Further research is needed for delayed hyperalgesia. Fifth, we only recorded patient satisfaction, but not surgeon satisfaction, based on our study protocol. However, there was no complaint in either group about the one surgeon. Finally, we did not conduct the analysis of cost and time saving; however, we saved about 1 USD (one syringe, one infusion set, and one 3-way stopcock) and 1 min by using a single syringe mixture and only setting the Schnider model for propofol but no Minto model for remifentanil, in each case in our institution. This was consistent with Malherbe and Barker reporting that the advantages of mixing propofol and remifentanil in the same syringe included a decreased cost of consumables, less medical waste, ease of delivery, and saving of time, especially during a high-turnover list with several-short duration procedures in quick succession [19].

## 5. Conclusions

In this study, the only significant difference between the two groups was TCI adjustment frequency. Accordingly, this RP mixture provided adequate hypnotic and analgesia effects under BIS and ANI monitoring and acceptable hemodynamic fluctuations in patients undergoing breast cancer surgery within 1 h. It is of value by providing an alternative approach for surgical procedures within 1 h, especially when multiple TCI systems are unavailable.

## Figures and Tables

**Figure 1 ijerph-20-02094-f001:**
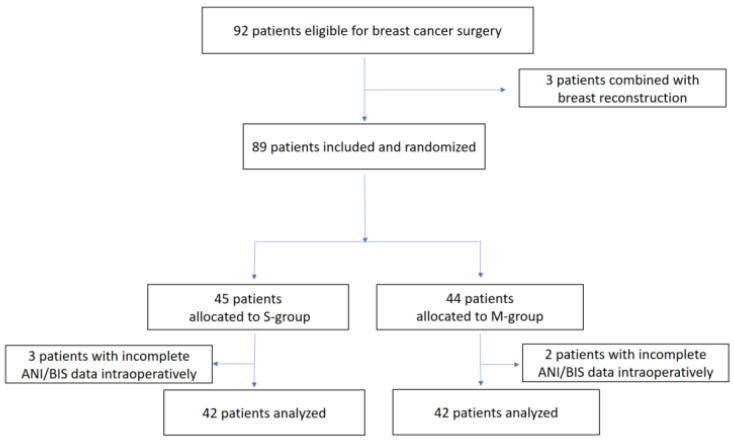
Flow diagram showing participants included in the analysis for the study.

**Figure 2 ijerph-20-02094-f002:**
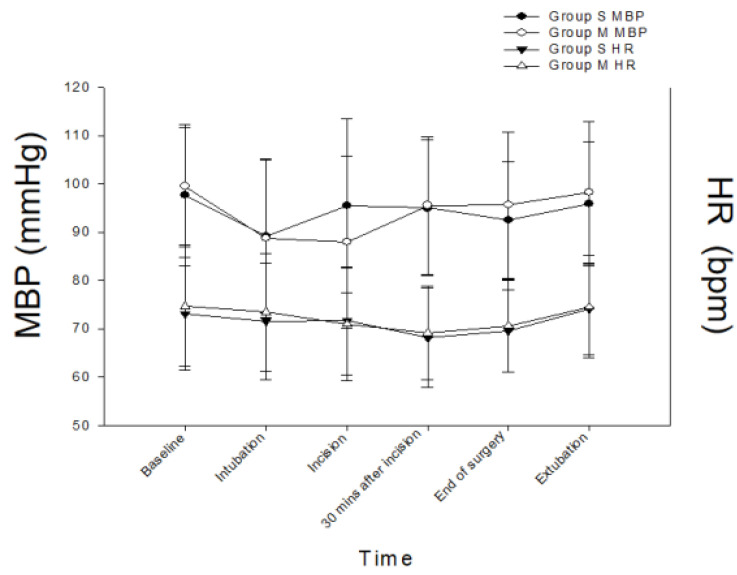
The hemodynamic parameters for both groups at baseline and during anesthetic course. S group: propofol and remifentanil delivered by separate TCI pumps; M group: mixed propofol/remifentanil delivered by a single TCI pump; MBP: mean blood pressure; HR: heart rate.

**Figure 3 ijerph-20-02094-f003:**
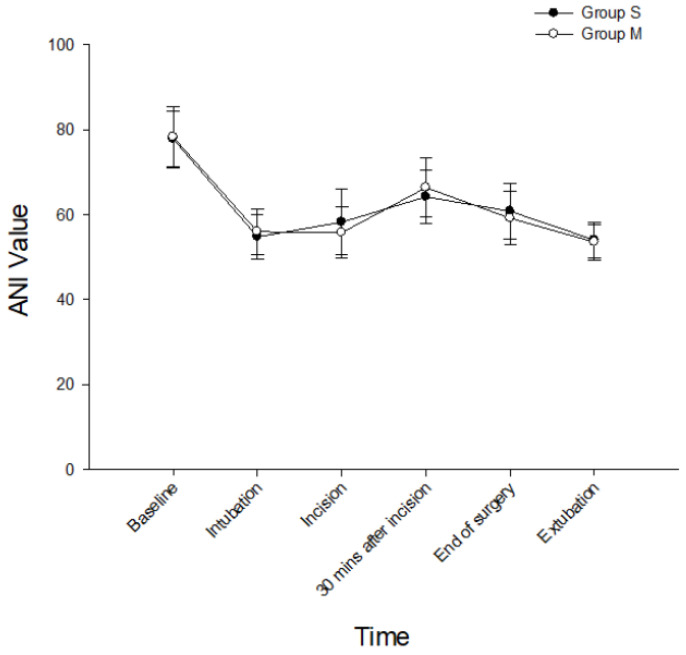
The ANI value for both groups at baseline and during anesthetic course. S group: propofol and remifentanil delivered by separate TCI pumps; M group: mixed propofol/remifentanil delivered by a single TCI pump; ANI: analgesia nociception index.

**Figure 4 ijerph-20-02094-f004:**
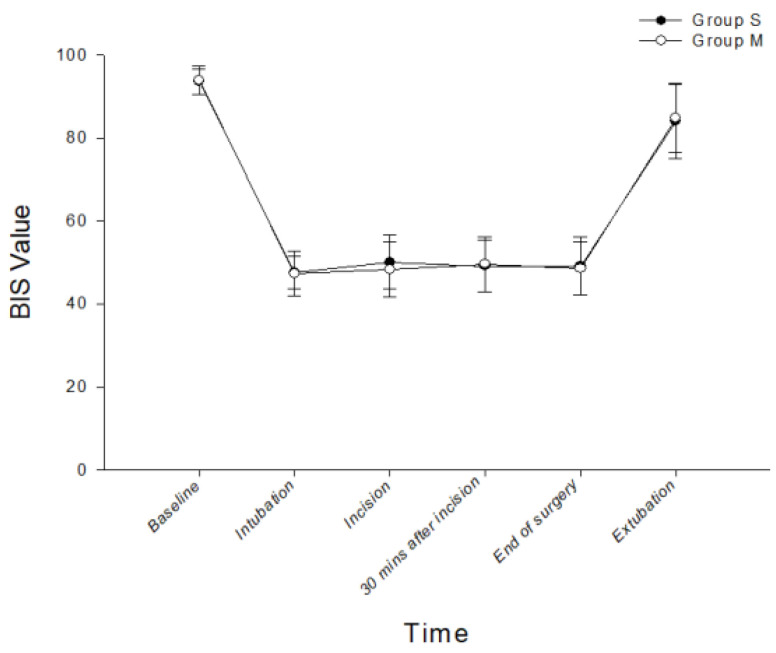
The BIS value for both groups at baseline and during anesthetic course. S group: propofol and remifentanil delivered by separate TCI pumps; M group: mixed propofol/remifentanil delivered by a single TCI pump; BIS: bispectral index.

**Table 1 ijerph-20-02094-t001:** Patients’ demographic characteristics, surgical procedures, and various time intervals.

Variables	S-Group (n = 42)	M-Group (n = 42)	*p* Value
Age (y/o)	55.4 ± 10.1	56.9 ± 11.5	0.520
Height (cm)	158.5 ± 6.4	156.4 ± 6.3	0.135
Weight (kg)	59.1 ± 10.3	59.4 ± 11.0	0.882
BMI (kg.m^−2^)	23.6 ± 4.3	24.3 ± 4.1	0.444
ASA class, II/III (n [%])	41/1 (97.6%/2.4%)	40/2 (95%/5%)	1
Comorbidities (n)			
Hypertension	9	9	1
Diabetes mellitus	3	6	0.483
Bronchial asthma	3	0	0.241
Thyroid disorder	3	4	1
Surgical procedures (n [%])			0.130
MRM	19(45%)	22 (52%)	
BCS	23 (55%)	20 (48%)	
Surgical time (min)	50.7 ± 10.4	51.73 ± 10.9	0.646
Anesthetic time (min)	67.4 ± 10.8	68.2 ± 10.1	0.715
Emergence time (min)	3.9 ± 2.7	2.9 ± 2.5	0.065

Data are expressed as mean ± standard deviation or number of cases (percentage) for categorical data. ASA, American Society of Anesthesiology; BCS, breast-conserving surgery; BMI, body mass index; MRM, modified radical mastectomy; N/A: not applicable.

**Table 2 ijerph-20-02094-t002:** Perioperative parameters of the two groups.

Variables	S-Group (n = 42)	M-Group (n = 42)	*p* Value
Frequency of TCI device adjustments (times)	3 (0–6)	2 (0–6)	0.005
Increase	1 (0–3)	1 (0–3)	0.068
Reduce Bolus (times)	2 (0–4)	1 (0–4)	0.016
Remifentanil	N/A	0 (0–1)	N/A
Propofol	N/A	0 (0–1)	N/A
Total consumption			
Propofol (mg)	437.0 ± 92.2	454.3 ± 87.6	0.380
Remifentanil (µg)	240.6 ± 71.9	228.5 ± 43.3	0.353
Postoperative analgesics rescue	12/42	14/42	0.814
(n [%])	(28.6%)	(33.3%)	
Patient satisfaction	4 (3–5)	4 (3–5)	0.610
PONV (n [%]) RIH (n [%])	0 (0%) 0 (0%)	1 (2.4%) 0 (0%)	1 1

Data are expressed as mean ± standard deviation, or medians with the range, or number of cases (percentage). N/A: not applicable; PONV: postoperative nausea and vomiting; RIH: remifentanil-induced hyperalgesia.

## Data Availability

The data presented in this study are available on request from the corresponding author.

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
