# Peer review of "Comparison of Single Target-Controlled Infusion Pump-Delivered Mixed Propofol and Remifentanil with Two Target-Controlled Infusion Pumps-Delivered Propofol and Remifentanil in Patients Undergoing Breast Cancer Surgery—A Prospective Study"

_ijerph, 2023, doi:10.3390/ijerph20032094_

Round 1
Reviewer 1 Report (New Reviewer)
The authors investigated the effectiveness of propofol and remifentanyl mixture in one syringe. My major concern is the compatibility of the drugs in this mixture. But the authors discussed this issue and the limitations really good in the discussion part. So I believe this paper would be a good guide for further investigations in this topic.
Author Response
Please see the attachment.

Reviewer 2 Report (New Reviewer)
Dear authors,
thank you for providing the present paper dealing with TCI application to a "fixed" mixture of propofol/remifentanil.
My concerns are general of nature. It seems very unclear, why the mixture without adequate modeling for the TCI should be preferred above two single pumps.
When pointing at your safety data, I wonder what these are? Your primary end-point used for the power calculation was changes of pump-flow rates, not safety. In addition, it is not clear how you assessed the safety.
The given data on BIS are certainly very close to each other, as this is your target to set your pumps.
While the introduction should have more information on the different used concentrations and models, the discussion is overloaded with these thoughts.
Please find more comments in the attached pdf file.

Author Response
Please see the attachment.

Reviewer 3 Report (Previous Reviewer 1)
All of previous comments have been satisfactorily dealt with in this revision. I have no more comments, and suggest this manuscript accept for publication.
Author Response
Please see the attachment.

This manuscript is a resubmission of an earlier submission. The following is a list of the peer review reports and author responses from that submission.
Round 1
Reviewer 1 Report
Manuscript (Manuscript ID: ijerph-1968820) titled: “Comparison of single target-controlled infusion pump delivered mixed propofol and remifentanil with two target-controlled infusion pumps delivered propofol and remifentanil in patients undergoing breast cancer surgery - a prospective pilot study” has been carefully reviewed. The authors performed a randomized clinical trial in a single hospital to assess the efficacy of remifentanyl-propofol mixture prospectively for TIVA in patients undergoing breast cancer surgery. In general, the paper is well written and the methods are largely appropriate. However, there are several ways in which the paper could be improved.
1. A pilot study, by definition, is a trial study carried out before a research design is finalized to assist in defining the research question or to test the feasibility, reliability and validity of the proposed study design. Usually, pilot studies should not be used to test hypotheses since the appropriate power and sample size are not calculated. In this study, study hypothesis has been created and sample size has been calculated. Therefore, I think the wording of “pilot study” could be removed.
2. Abstract: Conclusions should be drawn based on the results of the primary and secondary outcomes. I think those statements in lines 370-371 of the text, and are more adequate to state the conclusions of the study.
3. The confounding of anesthesiologist and surgeon factors should be considered, because experienced surgeons or anesthesiologists might affect the study outcomes. I suggest to adjust these factors before making conclusions.
4.Since authors show patient satisfaction in Table 2, I think the satisfaction of surgical team is also an important endpoint. So, I suggest the authors to add surgeons’ (or anesthetist nurses’ ) satisfaction in the result section and table 2, if possible.
5. Pleases make a summarize table of the clinical or laboratory studies use of an RP mixture for anaesthesia, it may help the readers to have a quick glimpse of the clinical application.
Reviewer 2 Report
I have read carefully the manuscript “Comparison of single target-controlled infusion pump delivered mixed propofol and remifentanil with two target-controlled infusion pumps delivered propofol and remifentanil in patients undergoing breast cancer surgery - a prospective pilot study”
The authors have administered propofol and remifentanil to adult patients (age around 50) undergoing breast surgery of about one hour duration, allocated to two different groups to receive the drugs either separately or mixed in the same syringe. They claim to have used target controlled infusions, but this is obviously not the case, specifically in the M group, even if they used TCI pumps.
The rationale for TCI administration is to use a pharmacokinetic model adapted to a given drug to convert a desired drug concentration (and thus drug effect) into a flow rate understandable by an infusion pump. This way, the predicted drug concentration achieved in the patient, and thus the effect, is stable and titratable. The authors have mixed into their syringe two drugs with widely different pharmacokinetic properties and administered the mixture guided by the pharmacokinetic model of only one of the drugs (propofol). This is pharmacologically absurd, and remifentanil in their mixture is not TCI administered, but the achieved remifentanil concentrations are unpredictable and not related to the patient’s needs. One of the major aims of anesthetic drug administration, specifically with hypnotic and antinociceptive monitoring such as used here, is to dissociate the control over hypnotic and analgesic needs. This is obviously impossible with the authors approach in their M group, as they themselves indicate “Third, we might meet a dilemma when the BIS and ANI indicate opposite directions for adjusting dosage, such as BIS < 40 and ANI < 50 or BIS > 60 and ANI > 70. This was the major weak point of our study in that we could not deal with separate infusion rates”. Moreover, if the effect obtained is without the desired boundaries, they correct their administration with manually administered boluses which are not “seen” by the infusion pump which will therefore estimate a completely wrong predicted concentration even with propofol.
The affirmation by the authors that “Because of the instability of the RP mixture, infusion rates decrease over time to maintain a stable blood concentration as the context-sensitive half-time rises; therefore, the rate of remifentanil infusion will decrease, resulting in lower concentrations that might not be enough for surgery” further confirms that they have not really understood what TCI is about. To my knowledge, provided the amount of remifentanil is diluted in a volume small enough not to disrupt the propofol emulsion (one more concern when mixing other drugs with propofol), the resulting mixture is fairly stable (Nicole Wylie, Lee Beale, Ian Westley; Consistency of remifentanil concentrations in propofol-Remifentanil infusions. A laboratory-based study; Paediatr Anaesth. 2022 Jun;32(6):727-731).
I strongly advocate them to read the following paper: Anthony R Absalom, Ann E Rigby-Jones, Andrew R Rushton, J Robert Sneyd. De-mystifying the "Mixifusor". Paediatr Anaesth. 2020 Dec;30(12):1292-1298
Given all this, why did they obtain clinically acceptable results?
They have recruited middle aged non obese patients, fairly fit (only 3 ASA class 3 patients in their whole population), who therefore have a wide pharmacodynamic tolerance to anesthesia, for rather short surgeries (about one hour). Also, apart from hypertension, the patients did not receive any cardiovascular drugs which might have impaired ANI measurements. Unfortunately, I cannot estimate the intraoperative course of anesthesia and specifically BIS and ANI evolution since no data were provided, not even the figures included in the paper.
The anesthetic regimen proposed seems hypnotic orientated with low remifentanil concentrations (2 ng/ml) in the S group which may blunt the remifentanil wandering concentrations in the M group.
To conclude: the authors have used a very controversial technique in patients with a wide tolerance to pharmacological errors for a short duration procedure. They claim that it is to reduce the work load and cost, but they monitor both the hypnotic and the antinociceptive effect … I fear that based on such papers clinicians might choose to go for different patients populations and different surgeries, without costly drug monitoring and encounter difficulties …
Reviewer 3 Report
Dear authors,
I have read your paper with large interest on a useful subject.
Here below I’m providing you with some (hopefully useful) suggestions to improve the quality of your work.
Wish you good luck for providing the revised version
Best regards
Reviewer
Introduction section
Introduction section may be more structured and motivational.
Page 2. Line 48.
Please modify procedure to procedural in “in procedure sedation”
Page 2. Line 80.
Please make sure that the final concentration and the preparation of the administered drugs are clearly explained in order to prevent errors in preparation of drugs. I think that 0.5 ml of remifentanil vial was added to 50 ml propofol 1% instead of 0.25 ml.
Page 4. Line 153.
Can the authors elaborate on how do they define and diagnose remifentanil induced hyperalgesia.
Page 4. Line 179.
The sample size calculation is based upon a difference of 0.6 points in the numerical VAS score that increases and decreases with one point at a time. How did the authors manage to get this point differences at 0.6? Please elaborate on this
Overall, this study is an attempt to verify the strategies that are commonly used in the authors’ institution. There is however a lack of significant benefit to the patients and even safety issues while preparing drugs and using pumps that are inappropriate for the TCI use of a propofol and remifentanil mixture.
The main aim of this work was to reduce the time-consuming process of preparation of two syringe pumps that are actually made and largely used based on PK-PD analysis for administration of propofol TCI and remifentanil TIVA. The authors tried to do this by using one syringe pump that is only appropriate for the use of propofol TCI. Instead, they changed the manufactory already prepared propofol 1% to a syringe of 50 ml and added presumably 0.5 ml of a manually prepared remifentanil from a concentration of 250 mcg per 0.25 ml to the propofol 1%. Thereafter they have used a TCI pump to administer this mixture. This pump is however, designed based on the PK-PD analysis for only propofol 1%. The co-workers were required to adjust the dose of these two drugs according to a complex protocol for a surgery that last less than one hour. Did the authors realize that all these actions are also time-consuming, and even a reason to make errors while preparing those medications and following their protocol. In addition, is it justifiable to use a syringe pump for something that is not yet studied and approved, where potentially patient’s harm and legal challenges may become an issue.
In my institution, this kind of surgeries are performed by using LMA and spontaneously breathing patients. Considering the experience of surgeons in their institution, why do the authors intubate these patients and administer NMBA repeatedly for this very short time frame? Please elaborate on this topic.
Reviewer 4 Report
Manuscript is prepared with proper study design and conclusion.
Reviewer 5 Report
This is well written paper and well executed research study.
My main concern is that in Conclusions: The major advantage of this modality is its convenience (less time consuming in both preparing drugs and setting TCI device ) and cost saving (less consumption of equipment) is not something that the authors were measuring in this study. Therefore, it should be deleted from the conclusion.
The only significant difference between two groups was adjustment frequency and that should be in the conclusions.
"The regimen was not inferior" should also be deleted , as the authors did not perform inferiority study.
The authors spent a lot of time and resources on this study. They should analyse their data and see how much money and time they saved by using single syringe mixture.
Round 2
Reviewer 3 Report
Dear authors,
Thank you very much for your responses.
I notice that some questions are answered by the text of same questions.
It seems that the study design clinically works at least in the authors’ institution, however, the study design of a research project should be scientifically based. The lack of proper PK-PD analysis for the use of a propofol- remifentanil mixture in a TCI pump that is specifically designed following intensive PK-PD analysis makes the results less desirable and reliable. Instead the authors could perform the same strategy and design by using TIVA pump and mg/ml/hr of drugs such as in other studies; eventually they may prefer to do PK-PD analysis first and then describe the dose strategy.
The concentration of remifentanil in the mixed group is very low only 5 mcg/ml. The authors stated that this was necessary for the stability of mixture. However, this has led to an anesthesia approach that in these patients was especially hypnotic oriented and not as such analgesic. What the authors describe as ‘Remifentanil-induced secondary hyperalgesia’ e.g. VAS > 4 following receiving a post-operative fentanyl bolus of 50 mcg (which is very low to my opinion) reflects purely the intensity of pain due to less attention to the analgesia intra-operatively.
Hopefully these suggestions would assist you further in scientific thoughts.
Best regards
Reviewer